# Microsatellite Instability Testing and Prognostic Implications in Colorectal Cancer

**DOI:** 10.3390/cancers16112005

**Published:** 2024-05-25

**Authors:** Vincent Ho, Liping Chung, Kate Wilkinson, Yafeng Ma, Tristan Rutland, Vivienne Lea, Stephanie H. Lim, Askar Abubakar, Weng Ng, Mark Lee, Tara L. Roberts, Therese M. Becker, Scott Mackenzie, Wei Chua, Cheok Soon Lee

**Affiliations:** 1School of Medicine, Western Sydney University, Penrith, NSW 2751, Australiayafeng.ma@unsw.edu.au (Y.M.); t.rutland@westernsydney.edu.au (T.R.); vivienne.lea@health.nsw.gov.au (V.L.); askar.abubakar@westernsydney.edu.au (A.A.); tara.roberts@westernsydney.edu.au (T.L.R.); therese.becker@inghaminstitute.org.au (T.M.B.); s.mackenzie@westernsydney.edu.au (S.M.); wei.chua@health.nsw.gov.au (W.C.); soon.lee@westernsydney.edu.au (C.S.L.); 2Ingham Institute for Applied Medical Research, Liverpool, NSW 2170, Australia; kate.wilkinson1@health.nsw.gov.au (K.W.); stephanie.lim@health.nsw.gov.au (S.H.L.); 3Department of Anatomical Pathology, Liverpool Hospital, Liverpool, NSW 2170, Australia; 4South Western Sydney Clinical School, University of New South Wales, Liverpool Hospital, Liverpool, NSW 2170, Australia; 5Macarthur Cancer Therapy Centre, Campbelltown Hospital, Campbelltown, NSW 2560, Australia; 6Department of Medical Oncology, Liverpool Hospital, Liverpool, NSW 2170, Australia; weng.ng@health.nsw.gov.au; 7Department of Radiation Oncology, Liverpool Hospital, Liverpool, NSW 2170, Australia; mark.lee2@health.nsw.gov.au; 8Department of Colorectal Surgery, Liverpool Hospital, Liverpool, NSW 2170, Australia; 9Discipline of Medical Oncology, School of Medicine, Western Sydney University, Liverpool, NSW 2170, Australia; 10Discipline of Pathology, School of Medicine, Western Sydney University, Campbelltown, NSW 2560, Australia

**Keywords:** colorectal cancer (CRC), microsatellite instability (MSI), mismatch repair (MMR), prognosis, primary site of tumors, formalin-fixed paraffin-embedded (FFPE) tissue samples

## Abstract

**Simple Summary:**

Microsatellite instability (MSI) is a hallmark of colorectal cancer (CRC) that is present in 10–15% of all patients with this condition. MSI results from the inactivation of the mismatch repair (MMR) pathway in tumors due to germline MMR mutations, known as Lynch syndrome or sporadic epigenetic gene silencing, leading to an increased mutation rate and genomic instability. In this study, we assessed the utility of the polymerase chain reaction (PCR)-based molecular method as an alternative to immunohistochemistry for determining MSI status in formalin-fixed paraffin-embedded tissues and investigated the clinicopathological significance and prognostic value of MSI in CRC. We found that patients with MSI-high (i.e., high MSI) had better overall and disease-free survival than those with microsatellite stability, and this result was significant in patients with right-sided CRC. Our results demonstrate that PCR-based MSI detection may serve as a useful prognostic predictor in patients with CRC and may have clinical value in the management of these patients.

**Abstract:**

Given the crucial predictive implications of microsatellite instability (MSI) in colorectal cancer (CRC), MSI screening is commonly performed in those with and at risk for CRC. Here, we compared results from immunohistochemistry (IHC) and the droplet digital PCR (ddPCR) MSI assay on formalin-fixed paraffin-embedded tumor samples from 48 patients who underwent surgery for colon and rectal cancer by calculating Cohen’s kappa measurement (*k*), revealing high agreement between the methods (*k* = 0.915). We performed Kaplan–Meier survival analyses and univariate and multivariate Cox regression to assess the prognostic significance of ddPCR-based MSI and to identify clinicopathological features associated with CRC outcome. Patients with MSI-high had better overall survival (OS; *p* = 0.038) and disease-free survival (DFS; *p* = 0.049) than those with microsatellite stability (MSS). When stratified by primary tumor location, right-sided CRC patients with MSI-high showed improved DFS, relative to those with MSS (*p* < 0.001), but left-sided CRC patients did not. In multivariate analyses, MSI-high was associated with improved OS (hazard ratio (HR) = 0.221, 95% confidence interval (CI): 0.026–0.870, *p* = 0.042), whereas the loss of DNA mismatch repair protein MutL homolog 1 (MLH1) expression was associated with worse OS (HR = 0.133, 95% CI: 0.001–1.152, *p* = 0.049). Our results suggest ddPCR is a promising tool for MSI detection. Given the opposing effects of MSI-high and MLH1 loss on OS, both ddPCR and IHC may be complementary for the prognostic assessment of CRC.

## 1. Introduction

Colorectal cancer (CRC) continues to be one of the most common cancers and a leading cause of cancer-related death. Although, in many patients, the precipitating factors that lead to CRC are unclear, approximately 10–15% of cases are associated with a molecular feature known as microsatellite instability (MSI). Microsatellites are short tandem-repeat sequences in the human genome that are often subject to replication errors, which are repaired by the mismatch repair (MMR) system machinery [1]. Patients with MSI are MMR-deficient due to the loss of expression of critical MMR proteins, such as MutL homolog 1 (MLH1), MutS homolog 2 (MSH2), MutS homolog 6 (MSH6), and PMS1 homolog 2 (PMS2). The loss of these proteins may result from aberrant epigenetic silencing or germline mutations of one allele, followed by somatic inactivation of the other allele in tumor tissue—a hereditary condition known as Lynch syndrome that is responsible for approximately 2–5% of CRC cases [2,3,4,5,6]. Both hereditary and sporadic loss of MMR lead to an elevated mutation rate and genomic instability, particularly at microsatellite regions of the genome. Consequently, MSI is among the most important hallmarks of CRC, and screening for MSI or MMR deficiency (dMMR) is recommended for those with a family history of certain cancers [7,8,9]. Although DNA repair deficiency often contributes to cancer development, it is also a potential prognostic marker, as the resulting tumors tend to be distinctively sensitive to DNA-damaging therapy [10].

CRCs with MSI are histopathologically distinct from microsatellite-stable tumors; they exhibit mucinous histology, poor differentiation, an increased number of tumor-infiltrating lymphocytes, and a Crohn’s-like lymphocytic reaction [11]. Moreover, MSI tumors more commonly arise on the right side of the colon, and the non-hereditary forms are more prevalent in women and more often located in the ascending colon [12,13]. Consistent with these unique clinical features, a number of studies have further found that MSI has prognostic value for CRC outcomes and response to treatment. In particular, results from several large population-based and retrospective cohorts and meta-analyses have shown that MSI is associated with improved survival in CRC patients [14,15,16,17,18], potentially resulting from elevated levels of immunogenic neoantigens and the consequent activation of cytotoxic T cells in tumors with MSI [19]. Others have found that MSI status may affect responses to certain cancer therapies, leading to improved responses to irinotecan or irinotecan-based chemotherapy and programmed cell death protein 1 (PD-1) blockade [20,21,22]. In contrast, several studies have reported no benefit from 5-FU therapy for patients with dMMR [23,24,25], whereas others detected benefits only for certain subtypes (e.g., some Stage III tumors and patients with germline MMR mutations) [15].

The utility of MSI as a risk factor for CRC occurrence, a prognostic indicator for outcome, and a predictive marker for response to treatment has led to the development of several different MSI detection methods. The most common gold standard approach involves the use of immunohistochemistry (IHC) to assess dMMR as a proxy for MSI [8,26]. For this method, tumor tissues are stained with antibodies against the primary MMR proteins, MLH1, MSH2, MSH6, and PMS2, and a lack of nuclear expression of any MMR protein is considered indicative of MMR loss. IHC can further indicate the likely identity of the mutated or inactivated MMR gene [27]. In contrast, PCR analysis of specific microsatellite loci can be performed to directly screen for MSI within the genome. The standard PCR-based approach recommended by the National Cancer Institute is based on the screening of five microsatellite markers—two mononucleotide repeats (*BAT26* and *BAT25*) and three dinucleotide repeats (*D2S123*, D5S346, and *D17S250*)—known as the Bethesda panel, in tumor and normal tissue [8,28]. Instability at a particular locus can be detected by the PCR analysis of fragment sizes through various methods, and samples are classified as MSI-high (MSI-H) if instability is detected at two or more loci. A second screening panel was developed based on five quasi-monomeric nucleotide repeats that are identical in size across individuals (*BAT-25*, *BAT-26*, *NR21*, *NR24*, and *NR27*), thus eliminating the need to analyze normal tissue in parallel [29,30]. Similar to the Bethesda panel, with this method, samples are classified as MSI-H if instability is detected at two or more loci. A commercial kit developed by Bio-Rad, which is based on the detection of these five loci by droplet digital PCR (ddPCR), was reported to show good agreement with IHC for the detection of MSI in endometrial cancer and CRC in a small proof-of-concept study [31].

In the present study, we sought to compare results from MSI screening using the Bio-Rad ddPCR MSI assay with those obtained using standard IHC in a selective-sample cohort of 48 CRC patients. We further performed Kaplan–Meier survival analyses and regression analyses to assess the clinicopathological significance and prognostic value of MSI in CRC. Our results indicate excellent agreement between the ddPCR MSI assay and IHC and reinforce the prognostic utility of MSI in CRC, particularly in patients with right-sided primary CRC.

## 2. Materials and Methods

### 2.1. Study Overview and Sample Selection

The study aimed to explore if the Bio-Rad ddPCR MSI assay could be a viable alternative to IHC for identifying MSI status in formalin-fixed paraffin-embedded (FFPE) tissue samples. It also sought to analyze the significance of MSI in CRC patients, looking at both clinic-pathological implications and prognostic value. A total of 48 FFPE tumor samples, obtained from patients diagnosed with stage I–IV CRC between 2000 and 2011 at Liverpool Hospital in Australia, were included in the study. The selection criteria prioritized including a higher number of MSI cases to enable a side-by-side comparison of IHC and ddPCR detection methods. Patients received radiotherapy, either an individual treatment of a 25 Gy dose in five fractions or a combination treatment of a 50.4 Gy dose in 28 fractions alongside 5-FU-based chemotherapy. Total mesorectal excision along with either anterior or abdominoperineal resection was performed to remove the tumors. Patients underwent follow-up through clinic appointments, blood tests, colonoscopy, and imaging based on the recommendation of the treating specialist. The research was approved by the South Western Sydney Local Health District Human Research Ethics Committee (Human Research Ethics Committee (HREC) Reference: HREC/14/LPOOL/186; project number 14/103), Sydney, Australia.

### 2.2. FFPE Tissue Processing and DNA Extraction

The FFPE tissue sections, varying between 5 and 10 µm in thickness with a tissue area not exceeding 2.25 cm^2^, were deparaffinized through immersion in xylene, followed by vigorous vortexing for 10 s and centrifugation at 20,000× *g* for 2 min at room temperature. DNA extraction from the deparaffinized tissue sections was performed using the QIAamp DNA FFPE Tissue Kit (QIAGEN, Hilden, Germany) following the provided instructions. The DNA obtained from FFPE tissue was quantified using the Qubit dsDNA High-Sensitivity Assay Kit (Thermo Fisher Scientific, Waltham, MA, USA). Prior to analysis, samples were made ready by combining 2 µL (~1 μg) of purified DNA with 198 µL of Qubit working solution to achieve a final volume of 200 µL. Subsequently, this mixture was left to incubate at room temperature for at least 2 min before quantification using the Qubit 4.0 fluorometer (Thermo Fisher Scientific).

### 2.3. Bio-Rad ddPCR MSI Assay

Following the manufacturer’s instructions, the Bio-Rad ddPCR™ MSI assay (Bio-Rad, Hercules, CA, USA) was conducted. Three separate ddPCR™ assays were established for the identification of five microsatellite markers (MSI assay 1 detecting *BAT25* and *BAT26*, MSI assay 2 targeting *NR21* and *NR24*, and MSI assay 3 for *Mono27*). Each sample was tested individually in wells containing both positive and non-template controls across all plates. The ddPCR mixture for each reaction comprised 1× ddPCR Multiplex Supermix for probes (Bio-Rad), 1× primer–probe mix, and 10–100 ng of tumor DNA extracted from FFPE samples, with a total volume of 22 µL. Droplets were created using the QX200 Droplet Generator (Bio-Rad) with 20 μL of the ddPCR mixture and 70 μL Droplet Generation Oil (Bio-Rad). Approximately 15,000 droplets were generated per well, with analysis including only samples resulting in >10,000 droplets.

The PCR reactions were carried out on a C1000 Touch Thermal Cycler (Bio-Rad) with the following protocol: initial denaturation at 95 °C for 10 min, followed by 40 cycles of denaturation at 94 °C for 30 s and extension at 55 °C for 1 min. The reactions concluded with a final incubation at 98 °C for 10 min, ramping up at a rate of 2 °C/s. Subsequent to PCR amplification, fluorescence signals were measured using the QX200 Droplet Reader (Bio-Rad), and the data were analyzed with the MSI10_FFPE.apfpack assay protocol files using the QX Manager Software (version 2.1.0.10) Premium Edition (Bio-Rad). Positive and negative controls were utilized to determine marker thresholds. MSI high (MSI-H) samples exhibited alterations in two or more microsatellite markers. MSI low (MSI-L) samples had alterations in one microsatellite marker, while samples with no alterations were categorized as having microsatellite stability (MSS). For statistical purposes, samples with MSI-L and MSS were grouped together in the MSS category.

### 2.4. IHC for MMR Proteins

The MMR proteins were immunohistochemically stained following the standard protocol of the Anatomical Pathology Laboratory at Liverpool Hospital. Prior to staining, tissue sections underwent a 1 h incubation at 60 °C. To prepare for staining, samples were deparaffinized with xylene, followed by a rehydration process through graded ethanol concentrations and ending with water. The sections were then treated with heated EnVision™ FLEX Target Retrieval Solution, High pH (DAKO, Glostrup, Hovedstaden, Denmark) in a hot water bath at 90 °C for 50 min, allowing them to cool at room temperature for an additional 20 min. The blocking of endogenous peroxidase was achieved with hydrogen peroxide for 20 min. Subsequently, slides were stained using primary monoclonal antibodies against MLH1 (mouse immunoglobulin, clone ES05, DAKO IR07961-2), MSH2 (mouse monoclonal, clone FE11, DAKO IR08561-2), MSH6 (mouse monoclonal, clone EP49, DAKO IR08661-2), and PMS2 (mouse monoclonal, clone EP51, DAKO IR08761-2). Each slide underwent a 10 min incubation with DAKO Mouse LINKER, followed by rinsing and a 30 min incubation with an anti-mouse secondary antibody. EnVision™ FLEX Substrate Working Solution (DAKO) was then applied for development on the slides. Consequently, the slides were stained with hematoxylin, rinsed with cold water, and immersed in Scott’s Bluing solution for 10 dips. After a quick rinse with cold water, the slides were dehydrated and mounted. Evaluation of MMR proteins relied on positive or negative nuclear staining for MLH1, MSH2, MSH6, and PMS2, without regard to the percentage of cells stained. The absence of staining for a specific MMR protein in a tumor indicated a loss of expression for that protein.

### 2.5. Statistical Analysis and Data Acquisition

The statistical analysis was conducted using Statistical Package for the Social Sciences (SPSS) for Windows version 29.0 (IBM Corporation, Armonk, NY, USA). Categorical variables were presented as the number of patients (%) in each group. Covariates included in regression analyses were sex, age, and tumor-node-metastasis (TNM) stage, differentiation, lymph nodes (LN) involvement, metastasis stage at diagnosis, lymphovascular and perineural invasion (LVI/PNI), primary tumor site, and adjuvant and neoadjuvant treatments. Clinicopathological variables in the MSI-H and MSS groups were compared using Pearson’s Chi-squared test. To measure the agreement between ddPCR and IHC results, Cohen’s kappa was utilized. The MSS group included both MSS and MSI-L samples for analysis. Kaplan–Meier survival analyses were conducted for the entire cohort and subgroups were categorized by primary tumor location. Univariate and multivariate analyses were carried out using Kaplan–Meier curves and Cox’s proportional hazard survival modeling. In all cases, *p* < 0.05 was considered statistically significant.

Data acquisition for mRNA expression profiles was performed for multiple genes in a single-study query via the cBioPortal for Cancer Genomics database (http://www.cBioPortal.org, accessed on 30 November 2023). We extracted the mRNA expression profiles of MMR genes, including *MLH1*, *MSH2*, *MSH6*, and *PMS2*. The log-rank *p*-values for Kaplan–Meier plots were generated to analyze the correlation between mRNA expression levels (i.e., loss of expression, altered group vs. normal expression, unaltered group) and patient survival. Loss of expression in any of these markers indicates a potential deficiency in the MMR system, which can lead to errors in DNA replication and repair, resulting in MSI. MSI status and primary tumor location of the samples were provided by two multi-center Atlas and Compass of Immune Cancer Microbiome (AC-ICAM) studies: one on colon cancer [32] and another on rectal cancer [33].

## 3. Results

### 3.1. Study Population

The clinicopathological characteristics of the 48 patients included in this study are detailed in Table 1. The median age was 67 years (range: 33–86 years), and the cohort comprised a similar percentage of male (54.2%) male and (45.8%) female patients. The tumor stage at diagnosis was T3/4 in most patients (77.1%); 43.7% of patients were LN-positive, 2.1% had metastatic disease, and 35.4% showed poor differentiation. LVI/PNI was detected in 20.8% of patients, and 54.2% received both neoadjuvant and adjuvant therapy. Within this cohort, 20 patients had right-sided CRC, and 28 had left-sided CRC. Patients were followed for a median period of 5.95 years (range: 0.59–11.68 years).

### 3.2. Concordance between Routine IHC and MSI Testing by ddPCR

In accordance with our sample selection criteria, results from the IHC staining of samples from our cohort of 48 patients indicated that 22 patients (45.8%) had MSI-H resulting from dMMR, whereas 26 (54.2%) patients expressed normal levels of MMR proteins. Ten cases of MSI were defined due to the absence of both MLH1 and PMS2 expression (Table 2). Because these proteins form a heterodimer, the altered expression of either the MLH1 or PMS2 protein due to germline mutation frequently leads to the absence of both. Figure 1 shows representative staining of tissue from patients with dMMR due to the loss of MLH1, MSH2, MSH6, or PMS2 expression. In dMMR patients, ten cases had a loss of both MLH1 and PMS2, three cases had a loss of both MSH2 and MSH6, three cases had PMS2 loss alone, five cases had MLH1 loss alone and one case had MSH6 loss alone. The patterns of dMMR expression are shown in Appendix A.

Molecular MSI testing using PCR-based approaches evaluates a specific panel of five monomorphic mononucleotide markers (i.e., *BAT25*, *BAT26*, *NR21*, *NR24*, and *Mono27*) to identify instability in these loci. Tumors are classified as MSI-H if two or more of the loci show instability, MSI-L if one of the markers shows instability, and MSS if none of the markers show instability. Here, using the Bio-Rad ddPCR assay, we found that 20 (41.6%) patient samples had MSI-H, three (6.3%) had MSI-L, and 25 (52.1%) had MSS (Table 2). MSS and MSI-L patients were combined under the MSS nominator for subsequent analysis. Figure 2 shows representative two-dimensional plots of ddPCR MSI assay results from four patients—one with MSI in 5/5 (100%) microsatellites (patient P4), two with MSI in 3/5 (60%) microsatellites (patients P9 and P19), and one with MSI in 1/5 (20%) microsatellites (patient P45)—based on the analysis of tumor DNA in FFPE tissue samples.

To calculate concordance between IHC and results from MSI molecular analysis by ddPCR, we performed an agreement measurement analysis using Cohen’s kappa (*k*) statistics; *k* values of 0.4–0.80 are considered moderate to good agreement, and *k* > 0.81 indicate strong agreement. We found that the results of MSI testing with the Bio-Rad ddPCR assay showed a high level of agreement with the results of MSI testing by IHC (*k* = 0.915). Moreover, relative to IHC as the gold standard, ddPCR exhibited a sensitivity of 90.9% and a specificity of 92.9% (Table 3).

### 3.3. Association between MSI Status and Clinicopathological Features

Next, we used the Pearson Chi-squared test to assess the significance of the relationship between MSI status and clinicopathological features (*p* < 0.05). Age, sex, tumor stage, metastasis, differentiation, and adjuvant and neoadjuvant treatment did not significantly differ between MSS and MSI-H patients, according to the analysis’s results (Table 4). On the other hand, we discovered that MSI-H was substantially correlated with LN involvement (*p* = 0.027) and LVI/PNI (*p* = 0.006). By comparing the MMR (MLH1, MSH2, MSH6, and PMS2) protein expression as determined by IHC and MSI status, we were also able to assess the state of the MMR pathway in patient samples. We found that MLH1 and PMS2 expression loss were significantly associated with MSI-H (*p* < 0.001 and *p* = 0.002, respectively; Table 4).

### 3.4. Prognostic Implications of MSI Status Determined by ddPCR in CRC

We then determined the association between MSI status by ddPCR-based detection and survival of patients with CRC. In the Kaplan–Meier survival analysis, we found that patients with MSI-H had better overall survival (OS; *p* = 0.038, Figure 3A) and disease-free survival (DFS; *p* = 0.049, Figure 3B) than patients with MSS, as determined by the log-rank test. We further investigated whether MSI status was associated with DFS in patients with differing primary tumor locations (i.e., right- vs. left-sided CRC). The results showed that, for patients with right-sided CRC, those with MSI-H had significantly higher mean DFS than patients with MSS (*p* < 0.001, Figure 3D); however, this association was not observed in those with left-sided CRC (*p* = 0.363, Figure 3C). Our results therefore suggest that MSI-H tumors have a better prognosis than MSS or MSI-L tumors, particularly for patients with right-sided primary CRC.

Through univariate analysis, we found that MSI-H was significantly associated with improved OS (hazard ratio [HR] = 0.324, 95% confidence interval [CI]: 0.105–0.998, *p* = 0.046; Table 5). Similarly, in multivariate Cox regression analysis, MSI-H was associated with better OS (HR = 0.221, 95% CI: 0.026–0.870, *p* = 0.042), whereas the loss of MLH1 expression was associated with worse OS (HR = 0.133, 95% CI 0.001–1.152, *p* = 0.049; Table 5). These findings suggest that MSI-H and loss of MLH1 expression together are strong prognostic factors for OS in patients with CRC.

### 3.5. Clinical Significance of MSI in CRC Patients

Lastly, to validate our findings, we analyzed RNA sequencing data from the cBioPortal for Cancer Genomics (http://www.cBioPortal.org, accessed on 30 November 2023) obtained from two individual cohorts of colon cancer [32] and rectal cancer patients [33]. For both cohorts, we generated Kaplan–Meier curves and calculated the log-rank *p*-values comparing MSI status, determined previously based on mRNA expression levels of MMR genes [32,33], and patient survival, as described in the Materials and Methods section. In the colon cancer cohort, Kaplan–Meier curves revealed that MSI-H patients showed a trend toward favorable survival probability (*n* = 348, *p* = 0.270; Figure 4A), but this association was not observed for those in the rectal cancer cohort (*n* = 373, *p* = 0.866; Figure 4B). MSI-H and MSS samples from the colon cancer cohort and rectal cancer cohort stratified based on primary tumor location are shown in Figure 4C and Figure 4D, respectively.

## 4. Discussion

MSI is a defining feature of a prevalent CRC subtype, which develops due to dMMR and has been associated with disease risk, prognosis, and response to treatment [5,34]. This condition can arise in those with an inherited MMR gene defect following the loss of the corresponding normal allele in tumor tissue or in others due to sporadic epigenetic silencing, resulting in the loss of MMR function and an accumulation of microsatellite mutations, resulting in MSI in the tumor [2,3,4,5,6]. From a diagnostic perspective, the loss of expression of a particular MMR protein can be detected using IHC, thereby enabling the identification of the mutated gene. Alternatively, dMMR can be identified by molecular methods that directly measure MSI at specific loci. In this study, we aimed to compare a commercial molecular ddPCR method for direct MSI detection with standard IHC, which is routinely used to assess the expression of key MMR proteins in tumors, in a selective sample cohort of 48 CRC patients treated at Liverpool Hospital, Sydney, Australia. The results from ddPCR showed strong agreement with the results obtained from IHC, with a kappa value of 0.915 and a sensitivity and specificity of 90.9% and 92.9%, respectively. In fact, only one of the 48 samples had differing MSI designations by IHC vs. ddPCR. Interestingly, this patient had locally advanced rectal cancer (T3N1), did not receive neoadjuvant therapy, and showed normal expression of all four MMR proteins in the FFPE sample by IHC, with alterations in 1/5 microsatellites by ddPCR. Accordingly, this patient was classified as MSI-L and grouped with MSS patients for survival analyses. We further assessed the prognostic significance of MSI by ddPCR-based detection in this cohort and found that, in Kaplan–Meier analyses, patients with MSI-H had better OS and DFS than those with MSS/MSI-L, as determined by the log-rank test. However, when patients were stratified by primary tumor location, only right-sided CRC patients with MSI-H showed significantly improved DFS relative to those with MSS. Moreover, multivariate analyses indicated that, although MSI-H was associated with significantly improved OS, the loss of MLH1 expression was significantly associated with worse OS. Extending our analyses to larger colon cancer and rectal cancer patient cohorts from published studies [32,33], we found that those in the colon cancer cohort with MSI-H trended toward a more favorable survival probability than MSS patients.

The Bio-Rad ddPCR MSI assay leverages a digital PCR system based on water–oil droplet technology, in which template molecules are amplified within nanoliter-sized droplets in 96-well plates [35], leading to smaller sample size requirements and reduced costs. After amplification, the droplets are analyzed using a customized droplet reader and QX Manager Software (version 2.1.0.10) Premium Edition to determine those that are positive and negative based on fluorescence measurements; Poisson statistics are then used to calculate the initial DNA template concentration in the sample [36]. In the present study, we found that this strategy showed strong concordance with the results from IHC. These results are in agreement with those from a small prior study evaluating the Bio-Rad ddPCR MSI assay, together with two other molecular MSI detection techniques, relative to IHC [31], suggesting the ddPCR assay holds value for clinical MSI assessment. This is supported by the results of a recent retrospective Chinese single-centre study [37] of 406 colorectal cancer patients, which showed a very high concordance rate between MMR IHC and MSI PCR analyses.

Consistent with prior studies, we also found that MSI-H was associated with improved OS and DFS [14,15,16,17,18]. However, when patients were grouped based on primary tumor location, an association between MSI-H and significantly improved DFS was only observed for those with right-sided CRC. This finding parallels results from a published report showing that MSI was associated with improved DFS in a cohort of 1009 CRC patients, although, in this case, the association was only significant in women [12]. Numerous studies have shown that tumors arising within the right (proximal) and left (distal) sides of the colorectum are histologically, molecularly, and clinically distinct, likely due to the differing physiological features of these two anatomic regions [38,39,40,41,42,43,44]. Notably, MSI-H is more commonly present in right-sided than in left-sided CRCs [13,38,45]. Somewhat paradoxically, however, overall outcomes for patients with right-sided CRCs are worse than those with left-sided CRCs [44]. Here, the specific reasons for the association between MSI-H and improved prognosis specifically for patients with right-sided CRC are unclear, although we posit that this reflects the inconsistent relationship between mortality and tumor location by stage due to tumor biology and, more specifically, MSI status, consistent with findings from other published work [46]. Given the limited scope of our study, this observation highlights the need for larger cohort studies directed at determining the prognostic value of MSI-H and dMMR in patients with left- vs. right-sided CRC.

We further note that, in multivariate analyses, although MSI-H was associated with improved survival, the specific loss of MLH1 expression was associated with significantly worse OS. Given that the epigenetic loss of *MLH1* is found in approximately 80% of sporadic MSI-H CRCs [47,48], our findings suggest differential prognoses for inherited vs. sporadic CRC arising due to MMR inactivation. This finding is consistent with recent studies that have suggested worse outcomes for CRC patients with sporadic vs. genetic MMR inactivation, possibly resulting from increased levels of neoantigens in genetic CRC, relative to sporadic CRC with dMMR [49,50]. However, larger studies will be needed to confirm these observations.

Our study has several limitations, the most notable of which is the small sample size, acknowledging a limited number of MSI patients from a single center. Because of the small sample, selection bias could not be avoided. Secondly, our study is limited by a lack of treatment data, including differential treatment assignments, which might confound results. The strengths of the study include the broadening of our findings from a small study group to two larger cohorts of patients with colon and rectal cancer. Although the results did not show statistical significance, there was a noticeable trend toward significance in the colon cancer cohort, which may have achieved significance had we been able to stratify this cohort according to tumor sidedness.

## 5. Conclusions

In conclusion, results from this study suggest that ddPCR is a promising tool for MSI detection. Our data further show that MSI-H is predictive for improved outcomes, particularly in those with right-sided CRC, whereas the specific loss of MLH1 is associated with worse outcomes. Given these findings, in accordance with current screening guidelines [26], both ddPCR (or similar MSI detection techniques) and IHC likely hold value for the prognostic assessment of CRC patients.

## Figures and Tables

**Figure 1 cancers-16-02005-f001:**
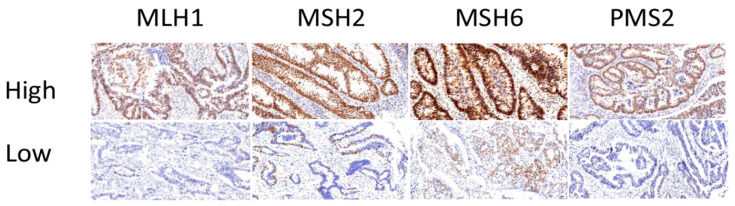
Representative immunohistochemical staining of MMR proteins in tumors. Expression of MutL homolog 1 (MLH1), MutS homolog 2 (MSH2), MutS homolog 6 (MSH6), and PMS1 homolog 2 (PMS2) in formalin-fixed paraffin-embedded (FFPE) tissue samples identified as normal expression (**top row**, labeled as high) and loss of staining (**bottom row**, labeled as low); magnification, 20×.

**Figure 2 cancers-16-02005-f002:**
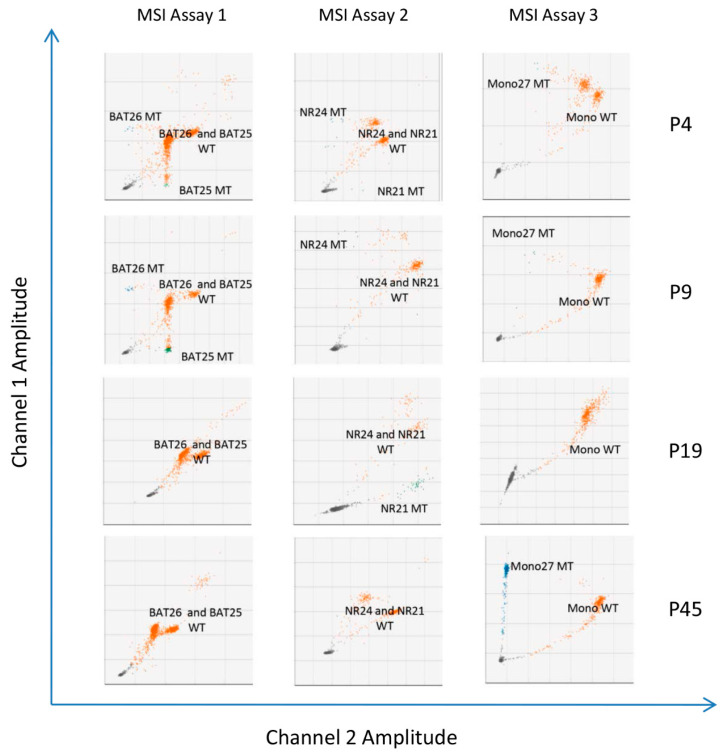
Two-dimensional droplet fluorescence plots from the three Bio-Rad ddPCR™ MSI assays (i.e., for *BAT25* and *BAT26*, *NR21* and *NR24*, and *Mono27*) performed on four patient samples (P4, P9, P19, and P45). Wild-type molecules are indicated by orange clusters, and unstable microsatellite molecules are indicated by blue clusters. The individual targets are shown on each plot.

**Figure 3 cancers-16-02005-f003:**
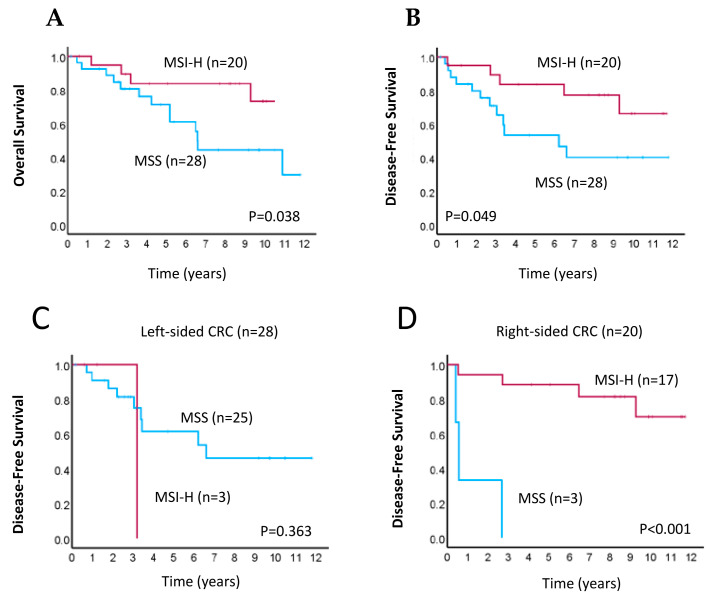
Kaplan–Meier survival analyses for CRC patients according to MSI status. (**A**,**B**) Kaplan–Meier curves of overall survival (**A**) and disease-free survival (**B**) in patients with MSI-high (MSI-H; pink line) or microsatellite-stable (MSS; blue line) tumors. (**C**,**D**) Kaplan–Meier curves of disease-free survival in patients with MSI-H (pink line) or MSS (blue line) tumors for those with left-sided (**C**) and right-sided (**D**) primary CRC.

**Figure 4 cancers-16-02005-f004:**
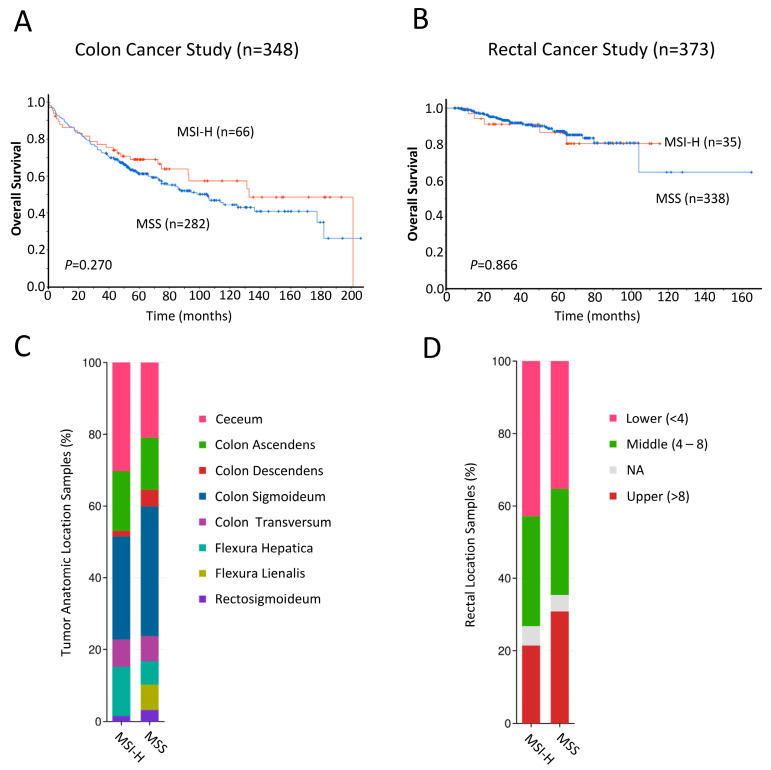
(**A**,**B**) Kaplan–Meier analyses comparing overall survival of n = 348 colon cancer (**A**) and n = 378 rectal cancer (**B**) patients from the cBioPortal for Cancer Genomics (http://www.cBioPortal.org, accessed on 30 November 2023 [31,32]) with MSI-high (red line) and MSS (blue line). Differences between groups were analyzed using the log-rank test, and *p*-values are shown. (**C**,**D**) Bar plot showing the proportions of MSI-H and MSS samples arising from various primary tumor locations in the colon cancer (**C**) and rectal cancer (**D**) cohorts.

**Table 1 cancers-16-02005-t001:** Clinicopathological characteristics of colorectal cancer (CRC) patients included in this study.

Clinicopathological Characteristics	Number of Patients
Age median/patients	67/48
<67	26 (54.2%)
≥67	22 (45.8%)
Sex	
Male	26 (54.2%)
Female	22 (45.8%)
Tumor stage	
T1, T2	11 (22.9%)
T3, T4	37 (77.1%)
Node stage	
N0	27 (56.3%)
N1, N2	21 (43.7%)
Metastasis stage	
M0	47 (97.9%)
M1	1 (2.1%)
Differentiation	
Well/moderate	31 (64.6%)
Poor	17 (35.4%)
LVI/PNI	
Absent	38 (79.2%)
Present	10 (20.8%)
Primary tumor site	
Right-sided	20 (41.7%)
Left-sided	28 (58.3%)
Treatment	
Neoadjuvant therapy	
No	22 (45.8%)
Yes	26 (54.2%)
Adjuvant therapy	
No	30 (62.5%)
Yes	18 (37.5%)

Abbreviations: LVI/PNI, lymphovascular and perineural invasion.

**Table 2 cancers-16-02005-t002:** Comparison between mismatch repair (MMR) and microsatellite instability (MSI) status determined by immunohistochemistry (IHC) and Bio-Rad droplet digital PCR (ddPCR), respectively.

Sample ID	MMR by IHC (Undetectable MMR Proteins)	MSI Testing by Bio-Rad ddPCR MSI Assay
P1	pMMR	MSS
P2	dMMR (MLH1, PMS2)	MSI-H
P3	dMMR (MLH1, PMS2)	MSI-H
P4	dMMR (MSH2, MSH6)	MSI-H
P5	pMMR	MSI-L
P6	pMMR	MSS
P7	pMMR	MSS
P8	pMMR	MSS
P9	dMMR (MLH1, PMS2)	MSI-H
P10	dMMR (MLH1, PMS2)	MSI-H
P11	dMMR (MLH1, PMS2)	MSI-H
P12	dMMR (PMS2)	MSI-H
P13	dMMR (MLH1, PMS2)	MSI-H
P14	pMMR	MSS
P15	dMMR (MLH1, PMS2)	MSI-H
P16	dMMR (MLH1, PMS2)	MSI-H
P17	pMMR	MSS
P18	dMMR (MLH1)	MSI-H
P19	dMMR (PMS2)	MSI-L
P20	dMMR (MLH1, PMS2)	MSI-H
P21	dMMR (MLH1, PMS2)	MSI-H
P22	pMMR	MSS
P23	dMMR (MSH2, MSH6)	MSI-H
P24	pMMR	MSS
P25	dMMR (MSH2, MSH6)	MSI-H
P26	pMMR	MSS
P27	pMMR	MSS
P28	pMMR	MSS
P29	dMMR (MLH1)	MSI-H
P30	dMMR (MLH1)	MSI-H
P31	dMMR (MLH1)	MSI-H
P32	dMMR (PMS2)	MSI-H
P33	dMMR (MLH1)	MSI-H
P34	pMMR	MSS
P35	pMMR	MSS
P36	pMMR	MSS
P37	pMMR	MSS
P38	pMMR	MSS
P39	pMMR	MSS
P40	pMMR	MSS
P41	pMMR	MSS
P42	pMMR	MSS
P43	pMMR	MSS
P44	pMMR	MSS
P45	dMMR (MSH6)	MSI-L
P46	pMMR	MSS
P47	pMMR	MSS
P48	pMMR	MSS

Abbreviations: dMMR, MMR deficiency; pMMR, MMR proficiency; MSI-H, microsatellite instability—high; MSI-L, microsatellite instability—low; MSS, microsatellite stable.

**Table 3 cancers-16-02005-t003:** Evaluation of the Bio-Rad ddPCR MSI assay relative to immunohistochemical staining for MMR proteins as the gold standard.

	Bio-Rad ddPCR MSI Assay
Kappa agreement measure (*k*)	0.915
*p*-value	<0.001
Sensitivity % (N)	90.9% (20/22)
Specificity % (N)	92.9% (26/28)

**Table 4 cancers-16-02005-t004:** Associations between MSI status determined by Bio-Rad ddPCR and clinical-histopathological features of CRC patients in our study cohort (*n* = 48).

	Clinical-Histopathological Features	MSI Testing *		
		MSS/MSI-L (%)	MSI-H (%)	*p*-Value
Sex	Male	69.2	30.8	0.096
	Female	45.5	54.5	
Age	<67	65.4	34.6	0.281
	≥67	50.0	50.0	
Tumor stage	T1/2	81.8	18.2	0.072
	T3/4	51.4	48.6	
Node stage	Negative	44.4	55.6	0.027
	Positive	76.2	23.8	
Metastasis stage	M0	57.4	42.6	0.393
	M1	100	0	
Differentiation	Well/moderate	67.7	32.3	0.074
	Poor	41.2	58.8	
LVI/PNI	Absent	68.4	31.6	0.006
	Present	20	80	
Adjuvant therapy	No	45.5	54.5	0.525
	Yes	55.6	44.4	
Neoadjuvant therapy	No	18.2	81.8	0.232
	Yes	92.3	7.7	
MLH1	Normal IHC	78.8	21.2	<0.001
	Loss of staining	13.3	86.7	
MSH2	Normal IHC	60.5	39.5	0.380
	Loss of staining	40	60	
MSH6	Normal IHC	62.8	37.2	0.066
	Loss of staining	20	80	
PMS2	Normal IHC	70.3	29.7	0.002
	Loss of staining	18.2	81.8	

Abbreviations: IHC, immunohistochemistry; LVI/PNI, lymphovascular and perineural invasion. * Bio-Rad ddPCR™ MSI assay.

**Table 5 cancers-16-02005-t005:** Univariate and multivariate Cox regression analyses evaluating the associations between clinical-histopathological features, including MSI status as determined by ddPCR, and overall survival.

		Univariate			Multivariate		
		HR	95% CI	*p*-Value	HR	95% CI	*p*-Value
MSI test	MSS	0.324	0.105–0.998	0.046	0.221	0.026–0.870	0.042
	MSI-H						
Sex	Male	1.190	0.445–3.115	0.723	3.126	0.296–32.99	0.343
	Female						
Age	<67	1.623	0.604–4.636	0.337	2.314	1.576–3.396	0.750
	≥67						
Tumor stage	T1/2	0.782	0.528–1.158	0.219	0.587	0.045–3.720	0.428
	T3/4						
Node stage	Negative	1.194	0.726–5.043	0.189	0.537	0.045–6.441	0.623
	Positive						
Differentiation	Well /moderate	1.073	0.396–2.906	0.890	0.954	0.904–27.981	0.964
	Poor						
LVI/PNI	Absent	0.797	0.228–2.787	0.722	1.956	0.050–75.891	0.719
	Present						
Adjuvant therapy	No	0.783	0.248–2.496	0.676	2.415	0.184–0.451	0.564
	Yes						
Neoadjuvant therapy	No	2.408	0.885–6.701	0.081	2.473	0.056–7.088	0.099
	Yes						
MLH1	Normal IHC	0.222	0.050–0.979	0.047	0.133	0.001–1.152	0.049
	Loss of staining						
MSH2	Normal IHC	0.466	0.061–3.588	0.464	0.478	0.082–4.719	0.878
	Loss of staining						
MSH6	Normal IHC	0.717	0.320–6.268	0.646	0.959	0.341–8.632	0.872
	Loss of staining						
PMS2	Normal IHC	0.666	0.190–2.344	0.527	0.744	0.530–9.702	0.490
	Loss of staining						

Abbreviations: HR, hazard ratio; CI, confidence interval.

## Data Availability

The data presented in this study are available on request from the corresponding author. The data are not publicly available due to the data’s size and due to privacy.

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
