# Peer review of "Microsatellite Instability Testing and Prognostic Implications in Colorectal Cancer"

_cancers, 2024, doi:10.3390/cancers16112005_

Round 1

Reviewer 1 Report

Comments and Suggestions for Authors

In this study, the author compared immunohistochemistry (IHC) with the droplet digital PCR (ddPCR) MSI assay among tumor samples from 48 colorectal cancer patients demonstrating high agreement between these two methods. They further assess prognostic significance of MSI to identify clinicopathological features associated with CRC outcome. They found that patients with MSI-high had better overall survival and disease-free survival than those with MSS. In addition, right-sided CRC patients with MSI-high remained with improved DFS relative to those with MSS while left-sided CRC patients did not. Furthermore, although MSI-high was associated with improved OS, loss of MLH1 expression was associated with worse OS. The authors thus recommended both ddPCR and IHC may be complementary for prognostic assessment of CRC.

There were several issues should be further clarified.

First, the authors should present the data about the rate of concordant loss of MLH1 and PMS2, MSH2 and MSH6.

Second, the authors should further analyze the rate of inherited subgroup (at least by Amsterdam criteria II) among the MLH1 loss patients.

Or third, the authors may presented the methylation status of these MLH1 loss tumors  

Reviewer 2 Report

Comments and Suggestions for Authors

It is not clear why so many authors for the study of only 49 samples. Particularly problematic is the number of authors that only provided "critical material". The number of this type of contributors has to be revised. 
Abstract and introduction are well written. Materials and methods also. Results are well presented. 
The value of this study is that shows that digital droplet PCR can be used for MSI detection in FFPE tissues. Although one must wonder will ddPCR ever replace the classical IHC in routine clinical pathology lab. Nevertheless, this is a valuable contribution to the general pool of knowledge. The other part of the study in which the authors perform clinicopathological correlations does not bring anything new, and due to small sample size authors did not manage to obtain the already known associations reported in the scientific literature. 

Reviewer 3 Report

Comments and Suggestions for Authors

Ho V et al. have made a good narrative review regarding microsatellite instability testing and prognostic implications in colorectal cancer. The manuscript is well written and I have no major criticisms. However, in order to enhance the comprehensiveness of this article, I would like to suggest the authors cite the similar studies from the Asian fellow researchers:

1. Asian Journal of Surgery. Volume 47, Issue 2, February 2024, Pages 959-967
